# Interactions between Cyanobacteria and Methane Processing Microbes Mitigate Methane Emissions from Rice Soils

**DOI:** 10.3390/microorganisms11122830

**Published:** 2023-11-21

**Authors:** Germán Pérez, Sascha M. B. Krause, Paul L. E. Bodelier, Marion Meima-Franke, Leonardo Pitombo, Pilar Irisarri

**Affiliations:** 1Department of Microbial Ecology, Netherlands Institute of Ecology (NIOO-KNAW), 6708 PB Wageningen, The Netherlands or gperez@fagro.edu.uy (G.P.); ssa@des.ecnu.edu.cn (S.M.B.K.); m.meima@nioo.knaw.nl (M.M.-F.); 2Laboratory of Microbiology, Department of Plant Biology, Agronomy Faculty, University of the Republic, Montevideo 12900, Uruguay; irisarri@fagro.edu.uy; 3School of Ecology and Environmental Sciences, East China Normal University, Shanghai 200062, China; 4Department of Environmental Sciences, Federal University of São Carlos (UFSCar), São Paulo 18052-780, Brazil; leonardopitombo@saneago.com.br

**Keywords:** carbon-cycling, methanotrophs, rice paddies, methanogens, agro-ecosystems, *Calothrix* sp., *Nostoc* sp.

## Abstract

Cyanobacteria play a relevant role in rice soils due to their contribution to soil fertility through nitrogen (N_2_) fixation and as a promising strategy to mitigate methane (CH_4_) emissions from these systems. However, information is still limited regarding the mechanisms of cyanobacterial modulation of CH_4_ cycling in rice soils. Here, we focused on the response of methane cycling microbial communities to inoculation with cyanobacteria in rice soils. We performed a microcosm study comprising rice soil inoculated with either of two cyanobacterial isolates (*Calothrix* sp. and *Nostoc* sp.) obtained from a rice paddy. Our results demonstrate that cyanobacterial inoculation reduced CH_4_ emissions by 20 times. Yet, the effect on CH_4_ cycling microbes differed for the cyanobacterial strains. Type Ia methanotrophs were stimulated by *Calothrix* sp. in the surface layer, while *Nostoc* sp. had the opposite effect. The overall *pmoA* transcripts of Type Ib methanotrophs were stimulated by *Nostoc*. Methanogens were not affected in the surface layer, while their abundance was reduced in the sub surface layer by the presence of *Nostoc* sp. Our results indicate that mitigation of methane emission from rice soils based on cyanobacterial inoculants depends on the proper pairing of cyanobacteria–methanotrophs and their respective traits.

## 1. Introduction

Rice is the most important food crop in the world, but its production is considered a major contributor to global warming [1] due to the emission of greenhouse gases (GHG) such as methane (CH_4_) and nitrous oxide (N_2_O) during different stages of growth or cultivation management [2]. For instance, the mean annual CH_4_ emission from rice paddies has been estimated to be around 24.9 Tg CH_4_ year^−1^ [3], with a ~20% contribution to global warming potential (GWP), while N_2_O emissions are contributing close to 5% [4].

CH_4_ emissions from rice soils are the result of the balance between production and consumption. Archaeal methanogens produce mostly CH_4_ under anoxic conditions during the flooding stage, which is the final step of organic matter decomposition in the soils [5]. Several enzymes are required to turn hydrogen and CO_2_ or acetate into CH_4_ [6]. One of the key enzymes in this process is the methyl coenzyme M reductase (MCR). The gene *mrcA*, responsible for catalyzing the α subunit of MCR, has been used as a molecular marker for studying these microorganisms in many ecosystems [7]. On the other hand, CH_4_ oxidation involves members from prokaryotes’, and the process can be performed under a range of O_2_ concentrations [8].

Denitrifying microorganisms from the NC10 phylum leads to CH_4_ oxidation under anoxic conditions [9]. Anaerobic methane oxidizing archaea (ANME) perform the process by using several electron acceptors such as NO_3_^−^, Fe^3+^, Mn^4+^, or humic acids [10] or in syntrophy with a bacterial counterpart as sulfate reducing bacteria [11]. Aerobic methane oxidizing bacteria (MOB) belong to the proteobacterial phylum, more specifically to the γ- (Types Ia, Ib, and Ic) and α- (Types IIa and IIb) classes [8,12]. This bacterial guild uses CH_4_ as a carbon and energy source, releasing CO_2_ as a product. In extreme environments (high temperature or low pH), methanotrophy is carried out by microorganisms of the phylum Verrucomicrobiota [12], known as Type III [8]. Non-canonical MOBs are also distributed in other phyla, such as Actinobacteria [13], Planctomycetota, and one putative Latescibacterota [14]. The first step of aerobic CH_4_ oxidation is catalyzed by methane monooxygenases, either in particulate or soluble forms [15]. The *pmoA* gene, which encodes for a subunit of the particulate form, is used for the detection of this group in several ecosystems [16]. Recent evidence has shown that Type I methanotrophs can be active under low O_2_ in the presence of Fe/Mn oxides [17] and SO_4_^2−^ [18], and these methanotrophs have genes related to denitrification or fermentation [19]. In rice fields, methanotrophic activity can mitigate 90% of CH_4_ emissions, with the highest activity within the first 10 cm of the soil [20,21]. This might be related to proteobacterial MOB activity, as they dominate surface rice soils [22]. However, the presence and activity of other methanotrophs as ANME in these systems have been acknowledged as significant in CH_4_ mitigation [23].

Another relevant microbial group in the upper layer of rice soils is oxygenic phototrophs, i.e., cyanobacteria, that have been extensively studied in these ecosystems [24]. These organisms can improve soil structure stability (synthesizing exopolysaccharides) and fertility by fixing nitrogen and carbon in these systems [25,26]. Most of the N-fixing cyanobacteria in rice soils belong to the *Nostoc* sp., *Calothrix* sp., and *Cylindrospermum* sp. genera (Nostocales order) [27]. Apart from their role in rice ecosystems, cyanobacteria have received attention for use in fuel production [28,29] and due to the impact of cyanobacterial blooms in lakes on carbon fluxes [30] by modulating methanogenic metabolic activity [31]. Another beneficial feature of these photosynthetic microorganisms is the fact that they can be involved in effective CH_4_ mitigation strategies, by stimulating methanotrophic bacteria. The phototroph–MOB interaction has been widely studied for biotechnological purposes, namely in CH_4_ removal in photoreactors using cyanobacteria [32] or microalgae [33]. Under these conditions, MOB would enhance their activity by interacting with phototrophs, turning this microbial interplay into a promising CH_4_ mitigation strategy. Yet, this syntrophic interaction between MOB and microbial phototrophs occurs naturally in aquatic [34] and terrestrial environments [35]. The commonly given explanation is that cyanobacteria CH_4_ mediated mitigation is supported by cyanobacteria derived O_2_, stimulating MOB, while MOB derived CO_2_ enhances the activity of cyanobacteria [36]. More recently, it was suggested that this interaction might be supported by the interchange of other metabolites, such as NH_4_^+^ and organic acids, rather than only by O_2_/CO_2_ [37]. In vitro incubations with cultured MOB and cyanobacteria demonstrated that cyanobacterial growth was supported by C coming from MOB [38]. However, how and where cyanobacteria modulate the abundance and activity of different groups of MOB (Types Ia, Ib, and II) with possible consequences for CH_4_ emissions in rice fields is not known [39]. In addition, it has not been explored whether the decrease in CH_4_ fluxes induced by cyanobacteria also involves the inhibition of methane producing microorganisms.

This study was performed with microcosms containing rice soil, in which two heterocyst-forming cyanobacterial isolates were used as inoculants. This was performed under flooded and controlled conditions, and the abundance (DNA level) and activity (RNA level) of methane cycling communities were analyzed in two physicochemical soil layers, namely surface soil (0–2 cm) and sub surface soil (2–4 cm). We hypothesized that (1) the two tested cyanobacterial isolates would mitigate CH_4_ emissions by increasing the abundance of various MOB subgroups (based on *pmoA* gene copies) and/or MOB activity (based on *pmoA* transcripts) or (2) by decreasing the abundance of methane producing microorganisms (based on *mrcA* gene copies). We further incorporated the role of habitat (soil layer) as an important driver of the response of methane cycling communities to cyanobacterial inoculation, expecting that cyanobacterial inoculation would homogenize physicochemical conditions in both layers in favor of MOB enhancing CH_4_ mitigation. 

## 2. Materials and Methods

### 2.1. Soil

Soil samples were taken from a rice field at the National Institute for Agricultural Research in Uruguay (33°15′ S, 54°10′ W) in March 2013. This soil is classified as Typic Argiudolls with pH = 5.7, 3.4% organic matter, 13 mg g^−1^, P Bray I, and 10 mg g^−1^ NO_3_^−^—N. Six sub-samples from the top layer (20 cm) of four random field plots were collected to make a composite sample. Before its use, the soil was air-dried, crushed, sieved (2 mm), and mixed thoroughly. Processed soil was kept at 25 °C in plastic containers until the beginning of the experiment (~three months), as described in [40]. 

### 2.2. Experimental Setup

A microcosm containing soil covered with a 4 cm-thick water layer was the experimental unit. This system simulated the flooding stage at which the biological processes of interest occur (Appendix A). Each microcosm (5 cm diameter × 15 cm height) consisted of a plastic container filled with 170 g of soil and covered on top with a water layer (distilled and autoclaved). The floodwater volume was kept constant by refilling each microcosm when necessary. Treatments comprised a control (without cyanobacterial inoculum) and two different cyanobacteria inoculates used separately (further details below). The experiment was designed with two factors (incubation time and inoculation treatments), while the microcosm position in the incubation room was randomized. Sampling time selection in this study followed the trends of CH_4_ fluxes described in the study system [41]. Briefly, CH_4_ production rates in Uruguay peak during flowering and decrease afterward. Rice is sown in dry soil, and the highest CH_4_ fluxes happen after 30–45 days of flooding when the rice-flowering stage takes place.

Two weeks before inoculation, microcosms with soil were flooded and left at 25 °C with a 50 µE m^−2^ s^−1^ photosynthetic photon flux density and a 16:8 light:dark period. This way, the physicochemical gradient between soil layers was created, and the microbial community would adapt to the experimental conditions [42]. On days 0, 15, and 30 days after inoculation, samples were taken from the water layer and two soil depths: surface (0–2 cm) and subsurface (2–4 cm). After taking the gas sample, microcosms were sacrificed for nutrient and nucleic acid extraction. Slurries from each layer were homogenized, suspended in RNALater^®^ (Ambion, Hamburg, Germany), flush frozen in liquid nitrogen, and kept at −80 °C until the nucleic acid extraction. For nutrient analysis, soil slurries were dried at 60 °C. Nutrient analyses in the floodwater were executed on water portions that were first filtered with Whatman GF/F (0.7 μm) and subsequently by nucleopore filters (0.2 μm) and kept frozen at −20 °C until the analysis was performed. On each sampling day, three experimental units from each treatment were analyzed and discarded (*n* = 3). 

### 2.3. Cyanobacterial Isolates and Culture Conditions

The two heterocyst-forming cyanobacteria, *Nostoc* sp. and *Calothrix* sp., used in this study were isolated from the same rice paddy field mentioned above [43]. These strains were chosen due to their high O_2_ photoevolution or N_2_ fixing activity [44]. They were grown in 250 mL Erlenmeyer flasks (150 mL culture) under constant shaking (~100 RPM) in BG11(0) medium [45] buffered with 10 mM HEPES, pH 7.6. Cultures grew under N-fixing conditions and were kept under the same light and photoperiod as described. Exponentially growing and homogenized cultures were used for the inoculation (~13 µg Chl *a* mL^−1^/microcosm) of the water column of the corresponding treatment. 

### 2.4. Determination of Soil and Water Physicochemical Parameters 

Ammonium (NH_4_^+^) and nitrate (NO_3_^−^) were extracted from a 5 g dry weight soil sample using a 2 M KCl solution in agitation for 30 min at 200 rpm. NH_4_^+^ content was determined with an improved Berthelot reaction [46] and NO_3_^−^ by its reduction using a Cd column [47]. In both cases, concentrations were determined by colorimetry. Organic carbon content (OC) was determined colorimetrically at 600 nm after wet oxidation with Kr_2_Cr_2_O_7_ in H_2_SO_4_ [48]. For pH determination, 2.5 g of dry weight soil was mixed with 10.5 mL of distilled H_2_O. 

### 2.5. Methane Fluxes

The system to determine CH_4_ fluxes consisted of a clamp perforated 1L lid glass jar with a septum (Appendix A). At each sampling date, each microcosm (*n* = 3) was placed in the jar, closed, and gas samples (10 mL) from the headspace were withdrawn using 25 mL syringes at 0, 30, and 60 min to calculate the fluxes of these gases. The samples were stored in evacuated 10 mL glass containers until analysis. The CH_4_ concentration was analyzed on a GC-2014 Shimadzu gas chromatograph (Tokyo, Japan) equipped with an FID detector and a Porapak Q column (L × O.D. × I.D.; 1.8 m × 3.125 mm × 2.1 mm) at 55 °C, with a gas carrier (N_2_) flow of 30 mL.min^−1^ and a flame ionization detector at 140 °C. 

### 2.6. Chlorophyll a (Chl a) Determination

To determine the cyanobacterial biomass added as an inoculant, the concentration of Chl a was determined by the Nusch method [49]. Briefly, pigment extraction was performed using 90% acetone, kept at −20 °C overnight (ON), and then quantified using a spectrophotometer.

### 2.7. Molecular Analyses

Nucleic acids were extracted based on Griffith and colleagues [50], with some modifications. Briefly, after RNALater^®^ removal, 0.5 g of soil was bead-beaten in a 2 mL vial filled with 0.5 g baked Zirconium beads (0.1 mm; Biospec Products Inc., Bartlesville, OK, USA) before 750 µL of NaHPO_4_ (120 mM; pH 8) and 250 µL of dodecyl sulfate were added. Once centrifuged, nucleic acids in the aqueous supernatant were extracted with equal volumes of phenol-chloroform-isoamyl alcohol (25:24:1 *v*/*v*/*v*) and afterward, chloroform-isoamyl alcohol (24:1 *v*/*v*) was used. Nucleic acid precipitation was performed using polyethylene glycol, centrifuged at 20,000× *g* at 4 °C for 30 min. Pellets were washed with 70% ethanol, then resuspended in RNAase-free water and stored at −80 °C. A portion of the extract was used for RNA preparation using the TURBO DNA-free kit (AMBION, AM1907) according to the manufacturer’s instructions. Finally, RNA transcripts were converted to cDNA using the Revert Aid First Strand cDNA synthesis kit (Thermo Fischer Scientific, Waltham, MA, USA). After each extraction procedure, 1% agarose gels stained with EtBr were run to check for nucleic acid degradation or genomic DNA contamination in RNA samples. Nucleic acid quantification was carried out by spectrophotometry using a nanodrop ND-1000 (Nanodrop Technology, Wilmington, DE, USA). 

### 2.8. Abundance of Methane Cycling Microorganisms 

The abundance of proteobacterial MOB (Types Ia, Ib, and II) and methanogens was estimated by qPCR of the *pmoA* (particulate methane monooxygenase) and *mcrA* (methyl coenzyme-M reductase) genes, respectively, with primer concentrations and PCR protocols summarized in Appendix A. The abundance of proteobacterial MOB was determined at all sampling dates, whereas that for methanogens was determined only at the beginning and end of the incubation. Each qPCR reaction (20 µL final volume) for both genes consisted of 10 µL 2× SensiFAST SYBR (BIOLENE, Alphen aan den Rijn, The Netherlands), 1 µL of forward and reverse primers (targeting *pmoA*) and 3.5 µL (targeting *mrcA*), 1 µL bovine serum albumin (5 µg.µL^−1^, Invitrogen, Breda, The Netherlands), and 1–2 µL DNA template. For the non-template controls, a 1–2 µL volume of DNase- and RNase-free water was used. DNA extracts were diluted (1:100) to perform each PCR reaction. Each qPCR assay from each DNA sample was performed in technical duplicates. Standard curves were obtained from a 10-fold serial dilution of a known amount of plasmid DNA fragment from pure cultures representing the target gene (107–101 *pmoA* gene copies and 10^8^–10^1^
*mcrA* gene copies). Amplification efficiencies ranged from 82 to 95.2%, with R^2^ values between 0.98 and 0.99. Amplicon specificity was checked by the melting curve and by running samples on a 1% agarose gel. The *pmoA* qPCR assay was performed with a Rotor-Gene 6000 thermal cycling system (Corbett Research, Eight Mile Plains, QLD, Australia), and for the *mrcA* assay, an iCycler IQ5 (Applied Biosystem, Carlsbad, CA, USA). 

### 2.9. Activity of the MOB

The activity of each MOB subgroup was studied using *pmoA* transcripts as a proxy. cDNA samples were diluted 1:10 to perform the PCR in the conditions described above.

### 2.10. Statistical Analysis and Data Visualization

All statistical analyses and plots were run and created in R (R Core Team version 4.2.0, 2022). The overall effect of cyanobacterial inoculation on MOB/methanogen abundance (*pmoA*/*mrcA* gene copies, respectively) and MOB activity (*pmoA* transcripts) was first assessed using linear mixed-effect models (LMMs) with R packages ‘lme4’ and ‘lmerTest’ (1.1-34). We used three models: (1) inoculation as a fixed effect and incubation period and soil layer as random effects (model 1); (2) incubation period as a fixed factor and inoculation and soil layer as random effects (model 2); and (3) soil layer as a fixed effect and inoculation and incubation period as random effects (model 3). When a fixed effect was significant (*p* < 0.05), means were compared using Tukey’s HSD test. In addition, a second strategy assessed the effect of cyanobacterial inoculation on MOB/methanogen abundance and MOB activity by soil layer separately. For this, we used a linear model to test the effect of inoculation and incubation period and their interaction (inoculation × incubation period) with the lm4 package. The same model was used to assess the effect of inoculation on CH_4_ fluxes and water and soil layers’ physicochemical parameters. 

Relationships between the abundance of methane cycling microorganisms at DNA and RNA levels and environmental variables were analyzed only on day 30 by Redundancy Analysis (RDA) using the “vegan” package (version 2.6-4). For this, the data were transformed using the decostand function before the analysis. qPCR data were Hellinger-transformed and environmental data were log-transformed. Explanatory variables were checked for non-collinearity. The best model was achieved by using the forward selection method. Pairwise associations among taxa/soil properties were calculated using the Pearson correlation method (1000 permutations). All plots were created using the ggplot2 package (version 3.4.3) and edited in Adobe Illustrator (v 27.6.1).

## 3. Results

### 3.1. CH_4_ Fluxes and the Effect of Cyanobacterial Inoculation

At the beginning of the experiment, all CH_4_ levels were negligible in all treatments, as shown in Table 1. Under our experimental conditions, we only found significant differences among treatments at the end of the incubation. In the control, CH_4_ fluxes were 20-fold higher in comparison to microcosms inoculated with *Nostoc* sp. (*p* = 0.02) and *Calothrix* sp. (*p* = 0.003). There was no significant difference between the effects of the two cyanobacterial inoculants on CH_4_ fluxes.

### 3.2. Effect of Cyanobacterial Inoculation on MOB Distribution, Abundance, and Activity

MOB subgroup distribution has followed a depth-related pattern since the beginning of the experiment (Appendix A). Types Ia and II were more abundant in the subsurface soil layer, while Type Ib was more abundant in the surface soil layer (*pmoA* gene copies). All MOB subgroups had higher *pmoA* transcripts in the surface soil layer. Type II MOB was the dominant MOB subgroup, followed by Type Ia in both soil layers, as supported by *pmoA* gene copy numbers. They were similar in terms of activity. Type Ib was the least abundant and active MOB subgroup. 

Cyanobacterial inoculation had a minor or no effect on the abundance or activity of MOB (Figure 1, Appendix A). Each cyanobacterium had differential effects on two MOB subgroups. For instance, the overall Type Ia abundance (*pmoA* gene copies) was not affected by inoculation. Yet, the significant effects of both cyanobacterial isolates on Type Ia abundance were seen in the surface layer on day 30 (Figure 1A, Appendix A). *Calothrix* sp. inoculation led to an increase in Type Ia toward the end of the experiment (Figure 1B). This 10% rise (Figure 1B) was also higher than the other treatments (*p* < 0.05). In the case of *Nostoc* sp. (Figure 1C), Type Ia decreased its abundance (~7%) in comparison to day 0. The control treatment did not show any significant change (Figure 1D). In the subsurface soil layer, cyanobacterial inoculation did not have any significant effect on Type Ia *pmoA* gene copies. 

The activity of Type Ia (*pmoA* transcripts) was not significantly affected under our experimental conditions (Figure 2A, Appendix A). Type II *pmoA* gene copies/transcripts did not show any significant change under the experimental conditions (Appendix A). Type Ib *pmoA* gene copies were not affected by inoculation, yet the overall Type Ib *pmoA* transcripts were significantly enhanced in the *Nostoc* sp. treatment in comparison to the control (Figure 2B). On the other hand, *Calothrix* sp. also led to an increase in Type Ib transcripts, yet it was not significantly different from the control. When assessed by soil layer, the inoculation effect on Type Ib transcripts was not significant.

### 3.3. Effect of Cyanobacterial Inoculation on the Abundance of Methane Producing Microorganisms

Methanogens were differentially distributed along the studied soil layers at the beginning of the experiment following the preincubation (Appendix A). Methanogens (*mrcA* gene copies) were around 40% more abundant in the subsurface soil layer than in the surface soil layer (*p* < 0.05). The overall abundance of this group was significantly affected by both cyanobacterial inoculants, leading to a decrease in *mrcA* gene copies (Figure 3A). When the inoculation effect on methanogens was analyzed by the soil layer, the response of the community differed. Methanogens from the surface layer did not show any significant change in their abundance (Figure 3B). On the contrary, methanogens from the subsurface soil layer of the *Nostoc* sp. treatment showed a significant decrease in comparison to the control (Figure 3C). We observed a similar trend in *Calothrix* sp., which, although not statistically significant, holds biological relevance when compared to the control group (*p* = 0.06).

### 3.4. Physicochemical Properties of Soil and Water and Effects on Methane Processing Microorganisms

Table 2 shows the fluctuations in soil physicochemical properties during the experiment. Water physicochemical variables are depicted in Appendix A. Descriptive statistics for all the studied compartments are found in Appendix A.

In the water column, nitrate concentrations were significantly influenced by time (*p* < 0.01), where concentrations on days 15 and 30 were higher than at time zero. pH was not affected by individual variables, but inoculation did have a significant impact in combination with incubation time (*p* < 0.01), leading to increased pH values over time in the inoculated treatments. *Calothrix* sp. inoculation had significantly increased pH by more than 1.5 units in comparison to the other treatments by day 15. The ammonium concentration had increased significantly towards the end of the incubation. 

In the soil layers, ammonium did not show any significant change over the course of the incubation. The nitrate content was significantly affected by the three factors tested. The nitrate became depleted in inoculated treatments, and pH significantly rose, reaching its highest value at the end of the experiment. In *Calothrix* sp., the OC content was significantly higher than the other treatments by day 30. 

We used RDA to assess the relationship between soil environmental factors and the abundance and activity of MOB and the abundance of methanogens on day 30. In the surface soil layer (Figure 4A), ammonium, nitrate, and organic content were the variables that better and significantly explained this relationship (*p* = 0.013; adj R^2^ = 0.66). RDA1 had eigenvalues of 68.2% (*p* = 0.008) and the RDA2 axis of 13.6% (*p* > 0.05), respectively. As shown in the biplot, OC content was more positively correlated with Type Ia *pmoA* gene copies. The second most important variable was nitrate (17.1%; *p* = 0.024), followed by ammonium (6.6%, *p* = 0.044). In the subsurface layer (Figure 4B), no combination of variables explained the variability in the abundance of methane cycling microbes in a significant way.

## 4. Discussion

In this microcosm study, cyanobacterial inoculation significantly decreased methane emissions from rice soils under flooded conditions, as previously reported [36,51]. More importantly, it provides new information on the effects of cyanobacterial inoculation on the dynamics of the methane cycling microbial communities in the early stages of rice soil flooding. Our results demonstrated that both cyanobacterial strains can mitigate CH_4_ emissions (first hypothesis) by modulating a subgroup of MOB and, to a lesser extent, the abundance of methanogens. We also showed that the effect of cyanobacterial inoculation and the response of CH_4_ cycling microorganisms vary with the cyanobacterial isolate applied and the soil layer where they live.

### 4.1. Inoculation Effect on MOB Abundance and Activity

Our study shows that each cyanobacterium had different effects on Type Ia and Type Ib MOB and on Type II. This is in line with previous reports that MOB subgroups hold different traits, thereby responding differently to perturbations (i.e., cyanobacterial inoculation) [20]. Type Ia MOBs are characterized as r-selected species (ruderals) responding to the availability of oxygen and methane faster than other MOB subgroups [52]. It has been suggested that interactions between MOB and phototrophs are modulated by oxygen [36]. Both cyanobacterial inoculants had similar oxygen photoevolution rates [44] and had opposite effects on Type Ia abundance (*pmoA* gene copies), which may suggest that this interaction is based on other cyanobacterial traits. Other studies have provided evidence that the mutual interchange of metabolites between MOB and cyanobacteria may also support their interaction [35,53,54]. The contrasting effect of *Nostoc* sp. and the indigenous phototrophic community in the control treatment with *Calothrix* sp. brings evidence that the MOB–phototroph interaction would be species specific. This is in line with evidence provided by a bioreactor for dissolved methane removal based on the syntrophy of cyanobacteria and MOB [53]. They showed that the preferential pairing of MOB with some cyanobacterial genera led to a decrease in MOB in the photogranule. Another aspect brought up by these authors is the role of the methylotrophic community in this interaction. MOB holds a tight interaction with methylotrophs [18,54], thus suggesting that the MOB–cyanobacteria interaction would be based not only on the metabolic interchange but also on the interaction with other members of the microbial community.

In our study, Type Ib clearly showed an environmental distribution different from the other MOB sub types. Type Ib has been described to be well adapted to rice paddies [55] and especially to the oxic–anoxic interfaces [56,57], as reflected in the higher Type Ib *pmoA* transcripts in the surface soil layer in our microcosms. Despite this fact, the increase in Type Ib activity due to *Nostoc* sp. inoculation could not be linked solely to the surface soil layer. This shows that the activity of Type Ib would not be depth-related, which also shows differences in life strategies with Type Ia MOB (depth-related response to inoculation). 

The lack of response to inoculation, growth dynamics, and soil distribution in Type II agrees with previous reports. Type II have slow growth rates (k-strategies), do not respond fast to shifts in their environment, and thrive better in nutrient-deprived environments [58] and low pH [59]. Another possibility could be related to the nitrate deprivation observed in the inoculated treatments. Some reports described that Type II activity was favored in conditions with high nitrate [60].

### 4.2. Effect of Cyanobacterial Inoculation on the Abundance of Methanogens

The methanogens had a clear distribution and contrasting responses to inoculation. This difference between soil layers may be a consequence of the preincubation that led the community in the surface layer to be more exposed to an oxic environment. The unresponsiveness of methanogens in the surface soil layer to cyanobacterial inoculation or the presence of an indigenous phototropic community (control) can be linked to an adaptation to oxygenation in the present community [61]. In this scenario, methane would have been actively produced in the subsurface soil layer. The effect of inoculation was more evident when considering the overall *mrcA* gene copies. However, methanogens from the subsurface were the ones affected by the inoculation. As reported before, a decline in *mrcA* gene copy numbers can be linked to a decrease in methane fluxes [62,63], which would explain the observed decline in methane emissions in the inoculated treatments. In our case, the decrease observed in *mrcA* gene copies took place only in the subsurface soil layer. Though the decreasing trend was marginally significant, it may seem to be the factor leading to the overall decrease in methanogen abundance. Similar cyanobacterial isolates as the ones used in this study showed that their biomass mineralization under controlled conditions can take 25 days [64]. This could indicate that, under our experimental conditions, cyanobacterial biomass was available for methanogens. Yet, previous studies reported that cyanobacterial biomass composition, as with algae, inhibits methanogenic activity [65] or that cyanobacterial biomass can favor the growth of other fermentative archaea rather than methanogens in lake sediments (0–5 cm) [66]. 

### 4.3. Effect of the Environment on the Distribution of Methane Cycling Microorganisms

Our experimental setup allowed the differential growth of methane cycling communities distributed in the two soil layers of flooded soils by a physicochemical gradient, as described previously [23,67]. In the surface soil layer, the abundance of Type Ia (*pmoA* gene copies) was strongly correlated to OC content. This is in line with observations that Type Ia MOB can use methane as well as other C sources [68] and that OC can enhance methane oxidation [69]. Nitrate has been positively linked to Type Ia growth [70,71]. In addition, a study also showed a positive correlation between Type Ia abundance and the ammonium content [72], as this MOB subgroup is better at assimilating N and thriving on the N pulse in comparison to Type II. 

In the subsurface soil layer, no correlations were found between environmental variables and MOB abundance. This may suggest that MOB communities were driven by other physicochemical parameters and that their composition and life strategies differed from those of the community in the surface soil layer. None of the measured environmental variables had a direct effect on the abundance of methanogens in any soil layer, yet other physicochemical variables such as the quality of organic C, P content, or soil Eh could hurt this community.

## 5. Conclusions

### Cyanobacteria as GHG Mitigators

The results obtained are in line with other studies reporting that cyanobacterial inoculation can be beneficial in reducing CH_4_ emissions in rice paddies. In addition to this, we show that this CH_4_ mitigation process driven by cyanobacteria affects MOB and methanogens differentially depending on the soil layer and the lifestyles of the methane microbial cycling community. Hence, these would suggest that one of the factors leading to the success of this mitigation strategy would be selecting the appropriate cyanobacterial strain (Figure 5). Based on the *pmoA* and *mrcA* gene dynamics in each treatment, the mechanisms by which cyanobacteria would exert a change on them would be a consequence of other traits than only photosynthetic oxygen production, as has been suggested. In this regard, further studies are required to disentangle the possible mechanisms describing how cyanobacteria steer the methane cycling community toward the mitigation of CH_4_ emissions in rice paddies.

It is worth mentioning that the primer sets used in the qPCR approach were focused on following the dynamics of aerobic proteobacteria methanotrophs and methanogens. Hence, we did not cover the potential effect of cyanobacterial inoculation on anaerobic methanotrophs in the NC1O phylum, or ANME, which may also be relevant in these systems. 

Despite the promising potential of cyanobacteria as CH_4_ mitigators, it is important to assess several aspects of “cyanobacterisation” [73] to achieve the desired effects in rice paddies. For instance, the inoculation formulation should comprise strains that promote the desired effect on MOB and methanogens and can be prepared on a large scale. Also, these cyanobacterial strains must be able to thrive under the type of soil and climatic conditions where the rice is grown. In this regard, global warming favors cyanobacterial blooms [74], and this can lead to the dominance of one or a few types of cyanobacteria, which may be toxin-producers, which can turn out to be an unexpected side effect of this mitigation strategy. Another factor to consider is that massive cyanobacterial growth before the rice seedling process may hinder rice production [75]. Also, this study did not include the effect of the rice plant itself, which plays an important role in the methane cycle in rice soils [76]. For instance, the plant may affect light penetration reaching the soil, hence affecting cyanobacterial growth, and would provide metabolites that would favor some metabolic pathways or also shape the methanotroph–phototroph interaction. Thus, CH_4_ mitigation by cyanobacteria would require further studies to elucidate the actual mechanisms that drive this microbial interaction and the applicability of this approach in the field, which may have an impact on the climate.

## Figures and Tables

**Figure 1 microorganisms-11-02830-f001:**
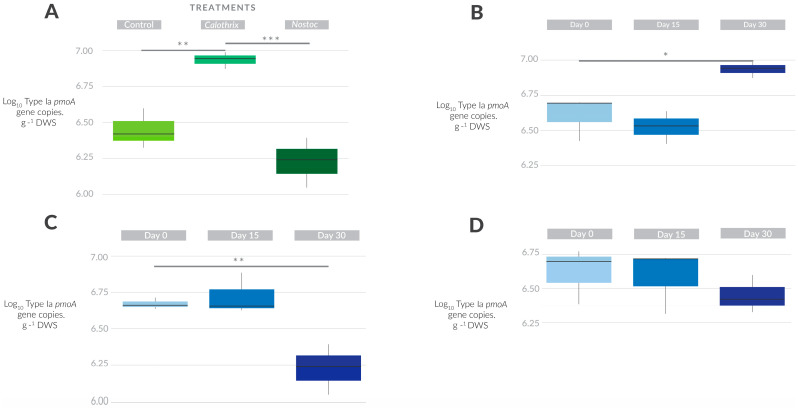
Effect of cyanobacterial inoculation on Type Ia abundance (*pmoA* gene copies) and activity (*pmoA* transcripts). (**A**) Dynamics of the *pmoA* gene copies from the surface soil layer along the incubation in the three treatments on day 30. (**B**) Abundance of Type Ia *pmoA* gene copies along the incubation of *Calothrix* sp. treatment in the surface soil layer on day 30. (**C**) Dynamics of *pmoA* gene copy from the surface soil layer along the incubation in the *Nostoc* sp. treatment. (**D**) Dynamics of the *pmoA* gene copy treatment from the surface soil layer along the incubation in the control. The vertical line within the box represents the median. The asterisks represent the *p* values: ‘***’ ≤ 0.001 ‘**’ ≤ 0.01 ‘*’ ≤ 0.05.

**Figure 2 microorganisms-11-02830-f002:**
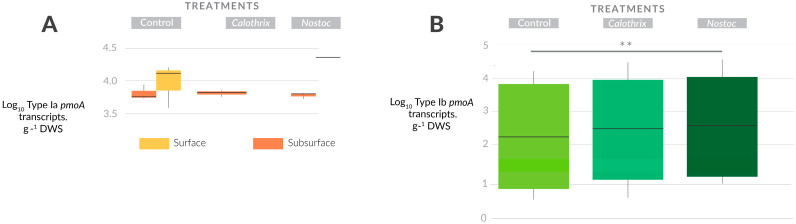
(**A**) Dynamics of Type Ia activity (*pmoA* transcripts) for the three treatments in both soil layers. (**B**) Overall Type Ib activity (*pmoA* transcripts) for the three treatments. The vertical line within the box represents the median. The asterisks represent the *p* values: ‘**’ ≤ 0.01.

**Figure 3 microorganisms-11-02830-f003:**
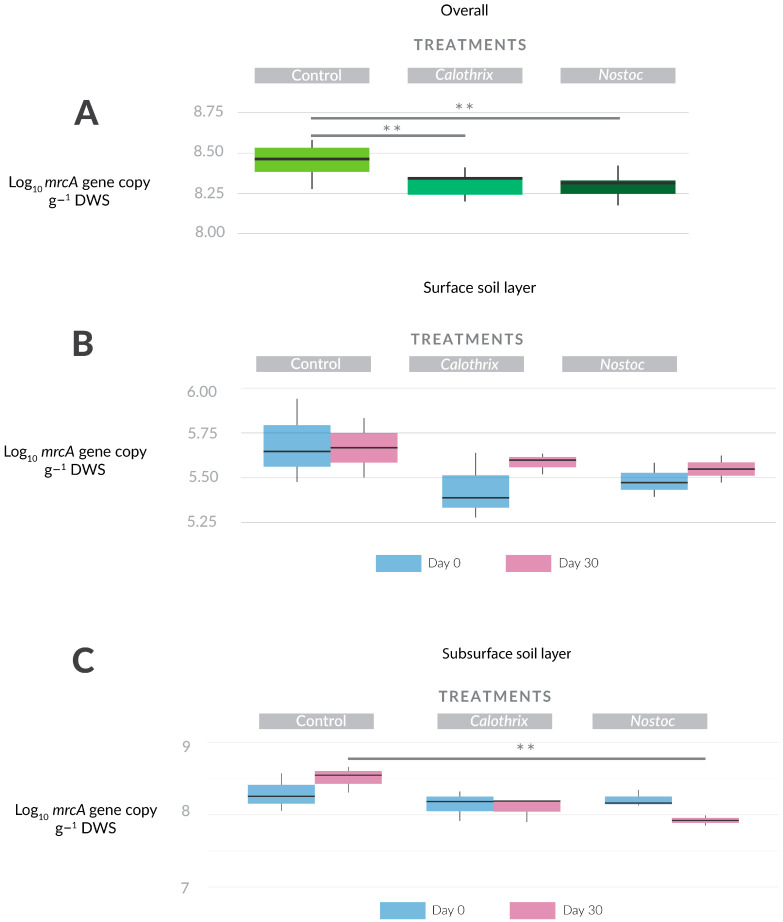
Effect of cyanobacterial inoculation on methanogen abundance (*mrcA* gene copies). (**A**) Overall *mrcA* gene copies per treatment. (**B**) Dynamics of *mrcA* gene copies on the surface soil layer. (**C**) Dynamics of *mrcA* gene copies on the surface soil layer. The vertical line within the box represents the median. The asterisks represent the *p* values: ‘**’ ≤ 0.01.

**Figure 4 microorganisms-11-02830-f004:**
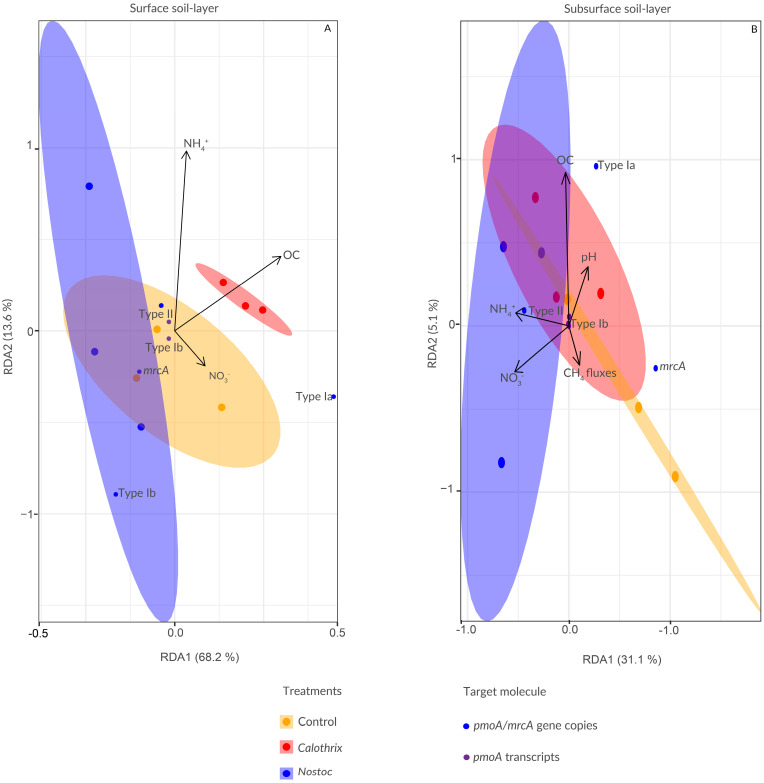
Redundancy analysis (RDA) plots show the influence of soil properties on methane cycling communities in both studied soil layers. The analyses were carried out for methane cycling abundance (*pmoA*/*mrcA* gene copy numbers and *pmoA* transcript number). Samples are colored by treatment. Vectors indicate the direction of increase for a given physicochemical soil variable, and their length depicts the strength of the relationship between the variable correlation and the ordination scores. Ellipses indicate 95% confidence intervals.

**Figure 5 microorganisms-11-02830-f005:**
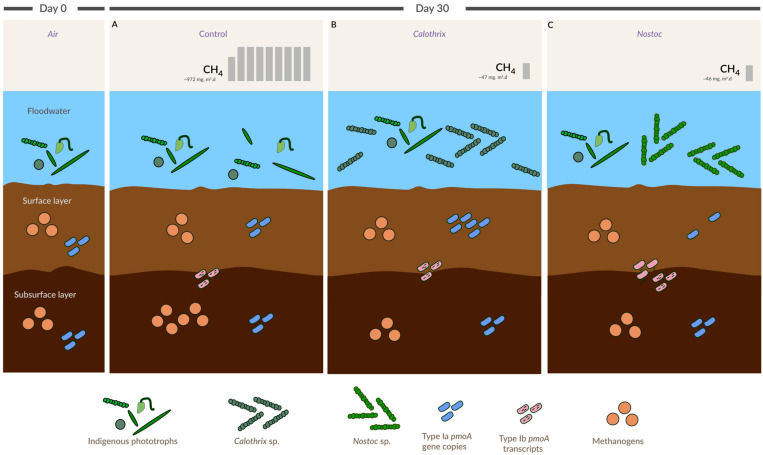
Overview of the main findings of the present study. (**A**) In the control treatment in our experiment, the indigenous microbial phototrophic community did not affect CH_4_ fluxes, and methanogens from the subsurface soil layer increased their abundance (*mrcA* gene copy numbers) after 30 days (in comparison to the abundance in the inoculated treatments). (**B**). In *Calothrix* sp.: inoculation reduced CH_4_ emissions (20 times in comparison to the control) and increased the abundance of Type Ia (surface soil layer) in comparison to other treatments at the end of the incubation. It also did not cause any fluctuation in the methanogenic abundance of any soil layer. (**C**) In *Nostoc* sp.: CH_4_ emissions dropped 20 times in comparison to the control treatment. The abundance of Type Ia (*pmoA* gene copies) in the surface soil layer was reduced after 30 days of incubation (in comparison to the previous sampling dates of this treatment). The overall activity of Type Ib *pmoA* transcripts increased in the *Nostoc* sp. treatment, whereas methanogenic abundance showed a decreasing trend only in the subsurface soil layer. The increase or decrease in MOB/methanogen cell numbers in the figure does not follow the actual trend observed. The phototrophic community representation is not at scale, shape, or color, which might not represent the real features of the organisms. Further details are in the main text.

**Table 1 microorganisms-11-02830-t001:** Methane fluxes in each treatment from three sampling dates. Values represent the mean ± standard deviation. Different lowercase letters show significant differences among days (*p* ≤ 0.05). Different uppercase letters indicate significant differences among treatments (*p* ≤ 0.05). “-” means below the detection limit.

Treatments	Incubation Period (d)	CH_4_ Fluxes (mg CH_4_·m^−2^·d^−1^)
Control	0	-
	15	3.62 ± 0.84 a
	30	1371.90 ± 565.61 bB
*Nostoc* sp.	0	-
	15	13.00 ± 18.55 a
	30	11.47 ± 5.70 aA
*Calothrix* sp.	0	-
	15	2.89 ± 1.05 a
	30	54.86 ± 74.82 aA

**Table 2 microorganisms-11-02830-t002:** Changes in soil and water column properties during the experiment. Values represent the mean ± standard deviation. Units of ammonium and nitrate are mg·N·kg^−1^ soil dry weight. OC stands for Organic Carbon expressed in %. B.D., below the detection limit, and (-) means “not determined”. Different lowercase letters on the same row show significant differences among days (*p* ≤ 0.05). Different uppercase letters in the same column of each variable indicate significant differences among treatments (*p* ≤ 0.05). The (*) and (°) next to each variable indicate that the treatment– incubation period interaction was significant (*p* ≤ 0.05).

Compartment							
		Surface Soil Layer			Subsurface Soil Layer		
		Incubation Period (d)					
Treatment	Variable	0	15	30	0	15	30
Control	pH * °	5.47 ± 0.07 a	5.41 ± 0.04 a	5.37 ± 0.0 b	5.30 ± 0.13 a	5.82 ± 0.08 b	5.94 ± 0.08 b
	NH_4_^+^	6.91 ± 3.43	3.91 ± 1.47	6.55 ± 0.27	5.82 ± 2.29	6.61 ± 2.2	7.19 ± 0.48
	NO_3_^−^ * °	0.65 ± 0.35 b	0.72 ± 0.27 b	0.50 ± 0.60 a	1.13 ± 1.06 b	0.41 ± 0.58 a	0.47 ± 0.61 a
	OC * °	2.06 ± 0.06 A	2.10 ± 0.04 A	2.03 ± 0.06 A	1.96 ± 0.04 Aa	2.09 ± 0.04 Aa	2.11 ± 0.02 Ab
*Nostoc* sp.	pH	5.46 ± 0.10 a	5.64 ± 0.22 a	5.78 ± 0.02 b	5.47 ± 0.15 a	5.76 ± 0.15 b	5.84 ± 0.02 b
	NH_4_^+^	6.81 ± 0.39	6.01 ± 4.03	6.96 ± 1.78	6.56 ± 1.72	7.85 ± 1.72	6.26 ± 3.87
	NO_3_^−^	1.14 ± 0.86 b	0.50 ± 0.40 b	0.01 ± 0.01 a	0.41 ± 0.10 b	0.19 ± 0.27 a	B.D.
	OC	2.07 ± 0.03 A	1.96 ± 0.03 A	2.15 ± 0.03 A	2.06 ± 0.10 Aa	1.95 ± 0.03 Aa	-
*Calothrix* sp.	pH	5.44 ± 0.02 a	5.67 ± 0.10 a	5.55 ± 0.10 b	5.56 ± 0.12 a	5.62 ± 0.19 b	5.50 ± 0.10 b
	NH_4_^+^	3.92 ± 0.02	5.29 ± 2.41	7.98 ± 1.16	4.66 ± 1.27	5.21 ± 1.40	6.26 ± 3.88
	NO_3_^−^	0.54 ± 0.23 b	0.70 ± 0.38 b	0.02 ± 0.02 a	0.15 ± 0.04 b	B.D.	B.D.
	OC	1.96 ± 0.04 B	2.32 ± 0.19 B	2.32 ± 0.20 B	2.00 ± 0.06 Ba	2.05 ± 0.16 Ba	2.58 ± 0.0 Bb

## Data Availability

The raw data supporting the conclusions of this article will be made available by the authors without undue reservation.

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
