# Peer review of "Interactions between Cyanobacteria and Methane Processing Microbes Mitigate Methane Emissions from Rice Soils"

_microorganisms, 2023, doi:10.3390/microorganisms11122830_

Round 1

Reviewer 1 Report

Comments and Suggestions for Authors

Reviewer comments

Manuscript: microorganisms-2691760 - Interactions between cyanobacteria and methane processing microbes mitigate methane emissions from rice soils.

The authors studied the response of methane cycling microbial communities to inoculation with cyanobacteria in rice soils. The authors performed a study comprising rice soil inoculated with either of two cyanobacterial (Calothrix sp. and Nostoc sp.) isolates obtained from a rice paddy. The results demonstrate that cyanobacterial inoculation reduced CH 4 emissions by 20 times. The effect on CH4 cycling microbes differed for the cyanobacterial strains. Type Ia methanotrophs were stimulated by Calothrix in the surface layer while Nostoc had the opposite effect. The overall pmoA transcripts of Type Ib methanotrophs were stimulated by Nostoc. Methanogens were not affected in the surface layer while their abundance was reduced in the subsurface layer by the presence of Nostoc. Their results indicate that mitigation of methane emission from rice soils based on cyanobacterial inoculants depend on the proper pairing of cyanobacteria-methanotrophs and their respective traits.

The data analysis methods are correct. There is control.

The English of the text is well written and well readable but needs additional checking with a professional translator.

The uniqueness of the text is more than 90% by AntiPlagiarism.NET.

The text contains some misspellings and typos. Also need to expand the part of the discussion.

There are some comments and questions:

1) What is novelty of this manuscript? Everything about interactions between cyanobacteria and methane processing microbes is known now. 

2) Line 106 - Typic Argiudoll - should be - Typic Argiudolls.

3) Line 117 - cyanobacterium inoculum - should be - cyanobacterial inoculum.

4) The Discussion part is weak, add more comparison your results with recent studies. For example: 

Khetkorn W, Raksajit W, Maneeruttanarungroj C, Lindblad P. Photobiohydrogen Production and Strategies for H2 Yield Improvements in Cyanobacteria. Adv Biochem Eng Biotechnol. 2023;183:253-279. doi: 10.1007/10_2023_216. Erratum in: Adv Biochem Eng Biotechnol. 2023;183:C1-C2. PMID: 37009974.

Costa DFA, Castro-Montoya JM, Harper K, Trevaskis L, Jackson EL, Quigley S. Algae as Feedstuff for Ruminants: A Focus on Single-Cell Species, Opportunistic Use of Algal By-Products and On-Site Production. Microorganisms. 2022 Nov 22;10(12):2313. doi: 10.3390/microorganisms10122313. PMID: 36557566; PMCID: PMC9786096.

Huang Y, Feng JC, Kong J, Sun L, Zhang M, Huang Y, Tang L, Zhang S, Yang Z. Community assemblages and species coexistence of prokaryotes controlled by local environmental heterogeneity in a cold seep water column. Sci Total Environ. 2023 Apr 10;868:161725. doi: 10.1016/j.scitotenv.2023.161725. Epub 2023 Jan 18. PMID: 36669671.

5) I have question. Soil was collected from of a rice field in March 2013. Why did you wait 10 years? Why didn't you collect samples this year? The ecology can be changed in 10 years.

6) The manuscript is important and interesting, well english written. The Introduction need in additional recent information, should be little extended. For example: 

Keller R, Goli K, Porter W, Alrabaa A, Jones JA. Cyanobacteria and Algal-Based Biological Life Support System (BLSS) and Planetary Surface Atmospheric Revitalizing Bioreactor Brief Concept Review. Life (Basel). 2023 Mar 17;13(3):816. doi: 10.3390/life13030816. PMID: 36983971; PMCID: PMC10057978.

Demirkaya C, Vadlamani A, Tervahauta T, Strous M, De la Hoz Siegler H. Autofermentation of alkaline cyanobacterial biomass to enable biorefinery approach. Biotechnol Biofuels Bioprod. 2023 Apr 8;16(1):62. doi: 10.1186/s13068-023-02311-5. PMID: 37029442; PMCID: PMC10082510.

Zhu Y, Chen X, Yang Y, Xie S. Impacts of cyanobacterial biomass and nitrate nitrogen on methanogens in eutrophic lakes. Sci Total Environ. 2022 Nov 20;848:157570. doi: 10.1016/j.scitotenv.2022.157570. Epub 2022 Jul 27. PMID: 35905968.

Zhou M, Zhou C, Peng Y, Jia R, Zhao W, Liang S, Xu X, Terada A, Wang G. Space-for-time substitution leads to carbon emission overestimation in eutrophic lakes. Environ Res. 2023 Feb 15;219:115175. doi: 10.1016/j.envres.2022.115175. Epub 2022 Dec 27. PMID: 36584848.

Please improve the manuscript according to the above comments.

Comments on the Quality of English Language

The quality of English is well. Moderate editing of English language required.

Author Response

Dear Reviewer,

Thank you very much for taking the time to review our manuscript. Please find the detailed responses below and the corresponding revisions/corrections highlighted/in track changes in the re-submitted file(s). We also inform you that a native speaker and scientist did the proofreading of the manuscript. Those changes appeared highlighted and with "track-changes", so you can notice these modifications. 

Regarding your comments:

  • What is novelty of this manuscript? Everything about interactions between cyanobacteria and methane processing microbes is known now.

Thank for your question. Yet, we respectfully disagree with your remark. We believe that there are still topics of the interaction that have not been covered. For instance, when doing a literature review on Google Scholar or Web of Science using the key words: methanotrophs + cyanobacteria + rice soils, both searching tools gave as first result the paper from Prasanna et al., 2002 as well as review papers citing this research. Hence, research on this topic in rice soils is scarce. In this regard, we believe that our experiment brought  some novel information on the interaction among cyanobacteria, methanotrophs and methanogens in rice flooded soils. In our research, we described the effect of cyanobacterial inoculation on the abundance and activity of the three MOB-subgroups (Ia, Ib and II) and how their microbial traits or habitat-specialization can modulate the interaction. On top of this, we addressed the research gap, studying the dynamics of both relevant microbial communities involved in the methane cycling: methanotrophs and methanogens. Another aspect that is worth mentioning is that our experiment setup with rice soil is different from a bioreactor or lakes where there is more information about this interaction. All in all, we believe that we are showing new results that will be interesting in this research field and may help to continue studying this microbial interaction.

Line 106 - Typic Argiudoll - should be - Typic Argiudolls.

Corrected, see line 136.

3) Line 117 - cyanobacterium inoculum - should be - cyanobacterial inoculum.

Corrected, see line 147.

4) The Discussion part is weak, add more comparison your results with recent studies. For example:

Thank you for the suggestions. As indicated, we incorporated some of these references in the version.

Khetkorn W, Raksajit W, Maneeruttanarungroj C, Lindblad P. Photobiohydrogen Production and Strategies for H2 Yield Improvements in Cyanobacteria. Adv Biochem Eng Biotechnol. 2023;183:253-279. doi: 10.1007/10_2023_216. Erratum in: Adv Biochem Eng Biotechnol. 2023;183:C1-C2. PMID: 37009974.

Added in line 76.

Costa DFA, Castro-Montoya JM, Harper K, Trevaskis L, Jackson EL, Quigley S. Algae as Feedstuff for Ruminants: A Focus on Single-Cell Species, Opportunistic Use of Algal By-Products and On-Site Production. Microorganisms. 2022 Nov 22;10(12):2313. doi: 10.3390/microorganisms10122313. PMID: 36557566; PMCID: PMC9786096.

Added in line 573.

Huang Y, Feng JC, Kong J, Sun L, Zhang M, Huang Y, Tang L, Zhang S, Yang Z. Community assemblages and species coexistence of prokaryotes controlled by local environmental heterogeneity in a cold seep water column. Sci Total Environ. 2023 Apr 10;868:161725. doi: 10.1016/j.scitotenv.2023.161725. Epub 2023 Jan 18. PMID: 36669671.

5) I have question. Soil was collected from of a rice field in March 2013. Why did you wait 10 years? Why didn't you collect samples this year? The ecology can be changed in 10 years.

Thank you for the question. Indeed, soil sampling, the experimental setup as well as the sample processing were performed in 2013. However, this manuscript was prepared in 2023.

6) The manuscript is important and interesting, well english written. The Introduction need in additional recent information, should be little extended. For example:

Thank you for your kind words on our work. We included pertinent information form these papers in the Introduction.

Keller R, Goli K, Porter W, Alrabaa A, Jones JA. Cyanobacteria and Algal-Based Biological Life Support System (BLSS) and Planetary Surface Atmospheric Revitalizing Bioreactor Brief Concept Review. Life (Basel). 2023 Mar 17;13(3):816. doi: 10.3390/life13030816. PMID: 36983971; PMCID: PMC10057978.

Demirkaya C, Vadlamani A, Tervahauta T, Strous M, De la Hoz Siegler H. Autofermentation of alkaline cyanobacterial biomass to enable biorefinery approach. Biotechnol Biofuels Bioprod. 2023 Apr 8;16(1):62. doi: 10.1186/s13068-023-02311-5. PMID: 37029442; PMCID: PMC10082510.

Added, see line 76.

Zhu Y, Chen X, Yang Y, Xie S. Impacts of cyanobacterial biomass and nitrate nitrogen on methanogens in eutrophic lakes. Sci Total Environ. 2022 Nov 20;848:157570. doi: 10.1016/j.scitotenv.2022.157570. Epub 2022 Jul 27. PMID: 35905968.

Added, see line 78.

Zhou M, Zhou C, Peng Y, Jia R, Zhao W, Liang S, Xu X, Terada A, Wang G. Space-for-time substitution leads to carbon emission overestimation in eutrophic lakes. Environ Res. 2023 Feb 15;219:115175. doi: 10.1016/j.envres.2022.115175. Epub 2022 Dec 27. PMID: 36584848.

Added, see line 77.

Please improve the manuscript according to the above comments.

The quality of English is well. Moderate editing of English language required.

As indicated before, the final proofreading was done a native speaker who is also a scientist. 

Reviewer 2 Report

Comments and Suggestions for Authors

In this study, the authors assess, via a microcosm study, the response of methane-cycling microbial communities to inoculation with two cyanobacterial isolates (Calothrix sp. and Nostoc sp.)

The authors used several methods to assess soil and water physicochemical parameters, pigments, and methane flux. And molecular approach to quantify Nucleic acids, and assess abundance of methane-cycling microorganisms and MOB activity.

The originality of the work lies in the study of the effects of cyanobacterial inoculation on the dynamics of the methane-cycling microbial communities in the early stage of rice soil  flooding.

Overall the work is well written. Despite this effort, the redaction of the paper contains some unclear points; and still requires amelioration in order to valorize this work.  

My comments are listed as follows:

Abstract:

Lines 18: replace “ two cyanobacterial (Calothrix sp. and Nostoc sp.) isolates” with “two cyanobacterial isolates (Calothrix sp. and Nostoc sp.) ”.

Line  21: Always add sp. to Calothrix and Nostoc throughout the text of the article (see lines  (27, 28, 238, 244-table, 268, 278, 295, 318, 365, 442 for Calothrix / 22, 28, 28, 139, 238, 260, 364, 445, 449 for Nostoc)

Materials and Methods

Line 104 : replace “collected “ with “sampled “

Line105 : delete “sampled “  

Line 109:  Why do you store the soil sample at 25°C? Preferably to conserve it at 4°C, in order to neutralize the action of the soil microflora that can deteriorate the quality of the sample. In this sense, did you carry out the physicochemical characterization of the soil before conservation or after?

Line 143: replace “BG110” with “BG11 (0) (without nitrogen source)

Overall, the manuscript can be accepted after minor revision.

Author Response

Dear Reviewer,

Thank you very much for taking the time to review our manuscript. Please find the detailed responses below and the corresponding revisions/corrections highlighted/in track changes in the re-submitted files.

Abstract:

Lines 18: replace “ two cyanobacterial (Calothrix sp. and Nostoc sp.) isolates” with “two cyanobacterial isolates (Calothrix sp. and Nostoc sp.) ”.

Done, see line 18.

 Line  21: Always add sp. to Calothrix and Nostoc throughout the text of the article (see lines  (27, 28, 238, 244-table, 268, 278, 295, 318, 365, 442 for Calothrix / 22, 28, 28, 139, 238, 260, 364, 445, 449 for Nostoc)

Done, see the same lines mentioned above.

Materials and Methods

Line 104 : replace “collected “ with “sampled “

Done, the sentences was reformulated. See lines 133-136.

Line105 : delete “sampled “ 

Done, see lines 133-136.

Line 109:  Why do you store the soil sample at 25°C? Preferably to conserve it at 4°C, in order to neutralize the action of the soil microflora that can deteriorate the quality of the sample. In this sense, did you carry out the physicochemical characterization of the soil before conservation or after?

Thank for your question and we understand your point. As cited in the text (ie: Murase, J.; Frenzel, P., 2008), Conrad and Klose, 2006, Hernandez et al., 2017 and many others, we followed this practice of storing soils at controlled room temperature. It was proved that the activity of these microorganisms was not affected. The soil physicochemical characterization that appears in sub-section 2.1 of Materials and Methods was done immediately after the soil sampling. Then, as described in the text, the analyses were performed on days 0, 15 and 30 under flooding conditions.

Line 143: replace “BG110” with “BG11 (0) (without nitrogen source)

There was a typo here. It was changed. See line 173.

Overall, the manuscript can be accepted after minor revision.